# Roles of CNC Transcription Factors NRF1 and NRF2 in Cancer

**DOI:** 10.3390/cancers13030541

**Published:** 2021-02-01

**Authors:** Hiroki Sekine, Hozumi Motohashi

**Affiliations:** Department of Gene Expression Regulation, Institute of Development, Aging and Cancer, Tohoku University, Sendai 980-8575, Japan; hozumim@med.tohoku.ac.jp

**Keywords:** CNC proteins, NRF1, NRF2, proteasome, *O*-GlcNAcylation

## Abstract

**Simple Summary:**

Although NRF1 (nuclear factor erythroid 2-like 1; NFE2L1) and NRF2 (nuclear factor erythroid 2-like 2; NFE2L2) belong to the CNC (cap‘n’collar) transcription factor family and share DNA recognition elements, their functions in vivo are substantially different. In cancer cells, while NRF2 confers therapeutic resistance via increasing antioxidant capacity and modulating glucose and glutamine metabolism, NRF1 confers therapeutic resistance via triggering proteasome bounce back response. Proteasome inhibition activates NRF1, and NRF1, in turn, activates the proteasome by inducing the transcriptional activation of proteasome subunit genes. One of the oncometabolites, UDP-GlcNAc (uridine diphosphate N-acetylglucosamine), has been found to be a key to the NRF1-mediated proteasome bounce back response. In this review, we introduce the roles of NRF1 in the cancer malignancy in comparison with NRF2.

**Abstract:**

Cancer cells exhibit unique metabolic features and take advantage of them to enhance their survival and proliferation. While the activation of NRF2 (nuclear factor erythroid 2-like 2; NFE2L2), a CNC (cap‘n’collar) family transcription factor, is effective for the prevention and alleviation of various diseases, NRF2 contributes to cancer malignancy by promoting aggressive tumorigenesis and conferring therapeutic resistance. NRF2-mediated metabolic reprogramming and increased antioxidant capacity underlie the malignant behaviors of NRF2-activated cancer cells. Another member of the CNC family, NRF1, plays a key role in the therapeutic resistance of cancers. Since NRF1 maintains proteasome activity by inducing proteasome subunit genes in response to proteasome inhibitors, NRF1 protects cancer cells from proteotoxicity induced by anticancer proteasome inhibitors. An important metabolite that activates NRF1 is UDP-GlcNAc (uridine diphosphate N-acetylglucosamine), which is abundantly generated in many cancer cells from glucose and glutamine via the hexosamine pathway. Thus, the metabolic signatures of cancer cells are closely related to the oncogenic and tumor-promoting functions of CNC family members. In this review, we provide a brief overview of NRF2-mediated cancer malignancy and elaborate on NRF1-mediated drug resistance affected by an oncometabolite UDP-GlcNAc.

## 1. Introduction

Cancer cells are subjected to various stresses, including oxidative stress and proteotoxic stress, which can interfere with cell growth [1,2]. Genetic, metabolic, and epigenetic alterations enable cancer cells to survive and proliferate in response to these stresses [1,2]. The KEAP1 (Kelch-like ECH-associated protein 1)–NRF2 (nuclear factor erythroid 2-like 2; NFE2L2) system is a molecular mechanism for the response to oxidative stress [3]. NRF2 is a member of the cap‘n’collar (CNC)–basic region-leucine zipper (bZIP) transcription factor family, which contains a well-conserved bZIP motif and CNC domain. NRF2 heterodimerizes with the small MAF (sMAF) protein, and the NRF2–sMAF heterodimer binds to a consensus sequence called antioxidant response element (ARE; GCNNN^G^/_C_TCA^C^/_T_), resulting in the activation of a battery of target genes and the subsequent elimination of oxidative stress [3]. In addition to the antioxidant function, NRF2 exerts anti-inflammatory function [4,5], stem cell regulation [6,7,8], and anti-aging function [9,10]. KEAP1 negatively regulates NRF2 by serving as a substrate recognition subunit of CUL3 (Cullin3)-based E3 ubiquitin ligase under normal conditions [11,12]. KEAP1 also serves as an oxidative stress sensor via its multiple redox-sensitive cysteine residues [13,14,15,16,17]. Reactive electrophile species attack specific cysteine residues and provoke structural changes in the KEAP1 protein, causing a functional decline of KEAP1–CUL3 E3 ubiquitin ligase and consequent stabilization of NRF2 [13,14,15,16,17]. In addition to the above-mentioned cytoprotective functions of the KEAP1–NRF2 axis, we and other groups have reported NRF2-dependent metabolic and epigenetic shifts in cancer cells [18,19,20,21]. Somatic mutations of the *KEAP1* gene, causing the constitutive activation of NRF2, were found among lung cancer patients and confer a robust growth advantage and tumor-initiating ability on cancers through the promotion of metabolic reprogramming and enhancement of cancer stemness [18,19,20,21,22,23].

NRF1 (nuclear factor erythroid 2-like 1, NFE2L1), another member of the CNC–bZIP transcription factor family, also heterodimerizes with sMAF, binds to AREs, and activates target genes. The NRF1–sMAF heterodimer is known to activate proteasome subunit genes via ARE sequences in response to proteasome inhibitors, which is referred to as the “proteasome bounce back response” [24,25]. As several cancers have and depend on increased proteasome activity, proteasome inhibitors are considered effective therapeutics for these cancers [26,27]. However, NRF1 antagonizes the effect of proteasome inhibitors on cancer cell growth through the proteasome bounce back response, suggesting that the suppression of NRF1 activity sensitizes cancer cells to these compounds. Recently, *O*-GlcNAcylation of NRF1 by *O*-linked N-acetylglucosamine (GlcNAc) transferase (OGT) was found to be indispensable for the nuclear accumulation of NRF1 and transcriptional activation of NRF1 [28,29]. UDP-GlcNAc, a substrate of the *O*-GlcNAc reaction, is synthesized from glucose and glutamine and is expected to be abundantly available in cancer cells that consume large amounts of glucose and glutamine [30]. Indeed, cells cultured under high-glucose conditions have increased cellular *O*-GlcNAcylated proteins and, as expected, increased NRF1 protein levels [28,29]. Consistently, the suppression of *O*-GlcNAcylation enhances the anticancer effect of proteasome inhibitors in xenograft experiments [28].

In this review, we first describe the distinct and redundant functions of NRF2 and NRF1 and the molecular mechanisms critical for their functions. Then, we introduce the NRF2 addiction of cancer cells, which confers unique features in terms of metabolism and epigenetic regulation, and the contribution of NRF1 to cancer malignancy, which is regulated by one of the oncometabolites, UDP-GlcNAc. In particular, we focused on the role of NRF1 in cancer cells as a transducer of metabolic signals to induce transcription.

## 2. Unique and Redundant Roles of NRF2 and NRF1 in Gene Expression

NRF2 and NRF1 were originally isolated as transcription factors related to NF-E2, and they possess highly conserved CNC domains and bZIP motifs [31,32]. Although the DNA-binding motifs of NRF2 and NRF1 are very similar to each other, the generation of gene-knockout mice revealed dramatic functional differences between *Nrf2* and *Nrf1* genes. *Nrf2*-null mice exhibited no obvious developmental defects [33] but exhibited an almost complete absence of inducible expression in response to oxidative stress [34], whereas *Nrf1*-null mice appear to be embryonic lethal due to impaired fetal liver erythropoiesis [35]. Although *Nrf1*-null mice exhibit no developmental abnormalities until embryonic day 12.5 [36], *Nrf1*:*Nrf2-*double-knockout embryos exhibit developmental delay at E10.5, suggesting the functional redundancy and compensation of NRF1 and NRF2 in early mouse development [28]. Mouse embryonic fibroblasts (MEFs) lacking both NRF1 and NRF2 exhibit higher levels of intracellular reactive oxygen species (ROS) than single knockout MEFs, suggesting a redundant and compensatory relationship between NRF1 and NRF2 in the control of cellular ROS levels [36].

In addition, NRF1 has been reported to activate proteasome subunit genes via AREs in response to proteasome inhibition [24,25]. Proteasome activity is required for the maintenance of normal neural function, and defects in its activity lead to neurodegenerative diseases [37]. Brain- or central nervous system-specific NRF1-knockout in mice show impaired proteasome function and the accumulation of ubiquitinated proteins, resulting in neurodegeneration [38,39]. We discuss later (Section 6) whether NRF2 is involved in the transcriptional regulation of proteasome subunit genes.

NRF2 enhances the cancer metabolic pathway through the gene expression involving the pentose phosphate pathway, purine nucleotide synthesis, glutaminolysis, and glutathione synthesis. Green boxes exhibit NRF2-target genes with ARE sequence. Red arrows show metabolic pathways activated by NRF2. R5P, Ribose 5-phosphate; G6P, Glucose-6-phosphate; F6P, Fructose 6-phosphate

## 3. Domain Structure of NRF2 and NRF1

Six domains are defined in NRF2 and NRF1, which are designated as NRF2–ECH homology (Neh) 1-6 domains, based on conserved amino acid sequences (Figure 1) [40]. The Neh1-6 domains are categorized into two groups according to their functions: common domains and unique domains. Neh1 belongs to a common domain that consists of a CNC region and bZip motif and mediates heterodimerization with sMAF proteins and DNA binding at ARE sequences. Neh6 is another common domain that is recognized by the β-TrCP-Cul1 E3 ubiquitin ligase complex and is categorized as second degron [41,42]. The remaining domains, Neh2, 3, 4, and 5, are regarded as unique domains. Neh2 is a degron of NRF2 containing two motifs, which are designated DLG and ETGE, and these interact with KEAP1 [3]. NRF1 also has completely conserved DLG and ETGE motifs in Neh2, which is consistent with a report describing that the NRF1 protein complex contains KEAP1 [43]. However, mutating the DLG or ETGE motifs does not change the transcriptional activity or protein level of NRF1 [44], indicating that the functional contribution of the Neh2 domain is different in NRF2 and NRF1. Neh3 and Neh4/5 of NRF2 serve as transcriptional activation domains [45,46,47,48]. In particular, Neh4/5 shows strong transcriptional activity upon interacting with many transcriptional activators, including histone acetyl transferases (p300 and CBP), ATP-dependent chromatin remodeler (BRG1), and the Mediator complex (MED16) [45,47,48]. Neh4/5 is also reported as a region that interacts with the glucocorticoid receptor (GR) [49,50]. GR recruits histone deacetylases (HDACs) through the Neh4/5 domain to the promoter or enhancer of NRF2 target genes in cells treated with both dexamethasone, a GR agonist, and diethyl maleate (DEM), an Nrf2 inducer, indicating that Neh4/5 is also essential for the repression of NRF2 activity [50]. Neh3 and Neh4/5 of NRF1 are also expected to serve as transcriptional activation domains. However, detailed analysis remains to be completed, because the NRF1 protein complex contains p300/CBP but not the Mediator complex [43]. In addition to Neh1-6 domains, NRF1 has a unique N-terminal extension that binds to the endoplasmic reticulum (ER), which endows NRF1 with responsiveness to ER stress [51].

## 4. Contributions of NRF2 and NRF1 to Cancer Malignancy

Cancer cells with the persistent activation of NRF2 rely heavily on NRF2 for proliferation, tumorigenesis, therapeutic resistance, and the maintenance of cancer stemness [18,21]. This NRF2 addiction status of cancer cells [52] is supported by potent antioxidant and detoxifying functions and unique metabolic adaptation [19,53] (Figure 2). In addition to a battery of cytoprotective genes, NRF2 directly activates genes encoding enzymes for the pentose phosphate pathway and serine synthesis pathway, resulting in the altered metabolic flux of glucose [18,20]. NRF2 also activates genes encoding a rate-limiting enzyme for glutathione synthesis, resulting in the remarkably enhanced production of glutathione, shifting the metabolic flux of glutamine to the glutathione synthesis pathway. An interesting metabolic feature of NRF2 addiction status is that glutamine consumption is accelerated because glutamate is actively used as a substrate for enhanced glutathione synthesis and because glutamate is excreted via the cystine–glutamate antiporter xCT, whose gene expression is also directly activated by NRF2. Thus, NRF2-activated cancers acquire potent antioxidant activity that leads to therapeutic resistance at the cost of vulnerability to glutamate restriction [19]. Additionally, epigenetic alterations are also responsible for the emergence of NRF2 addiction status by enhancing cancer stemness [21]. 

Intriguingly, NRF1 does not show an apparent contribution to metabolic regulation, although the consensus DNA-binding sequence of NRF1 is similar to that of NRF2. Conversely, NRF1 activity is under metabolic regulation in cancer cells. One of the oncometabolites, UDP-GlcNAc, turned out to be critical for NRF1 activation [28]. As a master regulator of proteasome subunit genes, NRF1 is critical for the “proteasomal bounce back response”, which confers resistance to proteasome inhibitors on cancer cells [28]. In the following sections, we focus on NRF1 as a regulator of proteasome activity and elaborate on the unique regulation of NRF1 activity by the oncometabolite UDP-GlcNAc for provoking the proteasome bounce back response.

## 5. NRF1 and Proteasome Regulation

### 5.1. O-GlcNAc Reaction

*O*-GlcNAcylation is a post-translational modification of nuclear and cytoplasmic proteins [54,55]. Enzymes catalyzing the addition and removal of the GlcNAc moiety are OGT and *O*-GlcNAcase (OGA; also known as MGEA5), respectively [56,57]. The addition of GlcNAc moieties to the hydroxyl groups of serine or threonine residues in target proteins by OGT requires a sugar nucleotide, UDP-GlcNAc, as a substrate [58]. UDP-GlcNAc is generated via the hexosamine biosynthetic pathway (HBP), to which glucose is shunted from the glycolytic pathway (Figure 3) [58]. In this pathway, glucosamine:fructose-6-phosphate aminotransferase (GFAT) catalyzes the conversion of fructose-6-phosphate, an intermediate of glycolysis, to glucosamine-6-phosphate by consuming glutamine [58]. Thus, cellular UDP-GlcNAc levels are affected by glucose and glutamine availability. As many cancer cells exhibit an elevated uptake of glucose and glutamine, UDP-GlcNAc production is most likely to be activated in cancers. Indeed, a recent comprehensive metabolomics-based study demonstrated that UDP-GlcNAc is abundant in colon cancers [59], providing justification for considering UDP-GlcNAc as an oncometabolite.

### 5.2. Role of O-GlcNAc Modification in Cancer

Protein *O*-GlcNAcylation is increased in many cancer cell types and is reported to associate with cancer malignancy [60,61]. In hepatocellular carcinoma (HCC) patients, high *O*-GlcNAcylation levels and low expression of OGA mRNA are associated with high frequency of HCC recurrence [62]. Gene expression analysis of urine sediments from bladder cancer (BC) patients revealed the relationship between tumor grades and OGT/OGA expression [63]. This study showed that OGA expression was significantly decreased in poorly differentiated BC (grade III) in comparison to grade I tumors. In contrast, OGT expression level was significantly higher in grade II/III BC than grade I BC, suggesting that cellular *O*-GlcNAcylation correlated with cancer malignancy. Other groups showed that OGT expression is involved in prostate cancer metastasis [64]. OGT mRNA is significantly increased in metastatic prostate cancers compared with primary cancers. Furthermore, high OGT expression is associated with poor disease-free survival after treatment with prostate cancer. These clinical data analysis demonstrated that increased cellular *O*-GlcNAcylation plays a key role in cancer progression and malignancy. 

Many oncogenic factors are *O*-GlcNAcylated and consequently acquire increased activity, promoting the proliferation of cancer cells [65,66,67,68,69,70,71,72,73]. For instance, *O*-GlcNAcylation of c-MYC at Thr58 blocks its phosphorylation, which hinders its recognition by the β-TrCP E3 ubiquitin ligase and blocked proteasomal degradation, resulting in the stabilization and increased expression of c-MYC target genes, including those encoding glycolytic enzymes [69,70]. β-Catenin is another example of a target of *O*-GlcNAcylation [71,72]. Mutations in and the overexpression of β-catenin underlie the progression of many tumors, including colon, liver, lung, breast, and ovarian tumors, where β-catenin serves as a structural component for cell adhesion and a regulator of transcription. Phosphorylated β-catenin is destabilized by β-TrCP-mediated ubiquitination and subsequent proteasomal degradation. Similar to c-MYC, *O*-GlcNAcylation of β-catenin inhibits GSK-3β-dependent phosphorylation, which is required for recognition by the β-TrCP E3 ubiquitin ligase, resulting in the high levels of accumulated β-catenin and target gene expression, such as E-cadherin expression [69,70,71,72]. Another example is HIF-1α, whose protein levels are controlled by cellular *O*-GlcNAcylation levels without being *O*-GlcNAcylated itself in breast cancers. The deficiency of OGT reduces alpha-ketoglutarate levels, which is a metabolite essential for the degradation of HIF-1α via its hydroxylation by HIF-PHDs, indicating that upregulated *O*-GlcNAcylation enhances HIF-1α activity and cell survival in cancer cells [73]. These results suggest that OGT may be a good therapeutic target for cancer treatment.

### 5.3. Molecular Mechanism for the Release of NRF1 from the Endoplasmic Reticulum

NRF1 is regulated at multiple levels, including post-translational modification and processing (Figure 4). In contrast to NRF2, NRF1 is initially synthesized as an ER-localized protein, and a large portion of N-glycosylated NRF1 exists in the ER lumen [51,74,75,76]. Under normal conditions, NRF1 is degraded through ER-associated degradation (ERAD) [24]. When cells are exposed to ER stress, including treatment with proteasome inhibitors, the C-terminus portion of NRF1 is retrotranslocated from the ER lumen to the cytoplasm by p97/VCP, which is followed by the de-N-glycosylation of NRF1 through NGLY1 [77,78]. De-N-glycosylated NRF1 is cleaved by DDI2, and processed NRF1 is translocated into the nucleus, where it heterodimerizes with sMAF before binding to ARE sequences to activate target genes, including those encoding proteasome subunits [79,80,81,82].

### 5.4. Molecular Mechanism of NRF1 Transactivation in the Nucleus

In addition to the regulation on ER, NRF1 protein levels are controlled after processing (Figure 5). In the nucleus, the human NRF1 protein is phosphorylated at Ser350 and Ser448/451 to enable its interaction with Fbw7 and β-TrCP ubiquitin E3 ligase, respectively, and then, it undergoes proteasomal degradation [41,83]. However, it remains unknown how nuclear NRF1 escapes phosphorylation-dependent proteasomal degradation and activates transcription. Comprehensive proteome analysis of the nuclear NRF1 complex revealed a regulatory mechanism of NRF1 in the nucleus. The OGT/HCF-1 complex was identified as an NRF1-binding partner [28,29,43]. Similar to NRF1-knockdown cells, deficiency of the OGT/HCF-1 complex attenuates the transcriptional activation of proteasome subunit genes induced by treatment with proteasome inhibitors [28]. In accordance with the reduced transcription of proteasome subunit genes, protein accumulation and chromatin binding of NRF1 were decreased in OGT/HCF-1-knockdown cells. It is now known that *O*-GlcNAcylation of NRF1 at Ser448/451 mediated by OGT/HCF-1 antagonizes phosphorylation and inhibits the interaction of NRF1 with β-TrCP, resulting in the accumulation of NRF1 in the nucleus and transactivation of its target genes [28].

### 5.5. NRF1 Regulation by Cellular UDP-GlcNAc through O-GlcNAcylation

Since OGT is essential for the accumulation and transcriptional activation of NRF1, the regulation of NRF1 activity is thought to be influenced by the availability of UDP-GlcNAc, which is a substrate of OGT for *O*-GlcNAcylation. Indeed, NRF1 accumulation responds to the cellular availability of UDP-GlcNAc. When cells are exposed to high glucose to increase cellular UDP-GlcNAc levels or treated with an OGA inhibitor to increase *O*-GlcNAcylated protein levels, NRF1 accumulation and cellular *O*-GlcNAcylation levels are increased. It is very plausible that NRF1 activity is regulated by UDP-GlcNAc synthesis, which is often enhanced in cancer cells [28].

### 5.6. The Proteasome as a Target for Cancer Therapy

The ubiquitin–proteasome system is related to many biological processes and maintains protein homeostasis. The proteasome is a 26S multiprotein complex that degrades polyubiquitinated proteins and is constituted by two subcomplexes called 20S, which has proteolytic activity, and 19S, which has regulatory activity, including deubiquitinating activity. An impaired proteasome system contributes to various diseases, such as neurodegenerative diseases, whereas the proliferation and survival of certain cancer species are known to be based on their elevated proteasome activity [84]. Proteasome inhibitors are hence considered promising drugs for the treatment of proteasome-activated cancers and are indeed used for patients with multiple myeloma or mantle cell lymphoma but not for those with solid tumors [84]. One of the main causes for the limited use of proteasome inhibitors is the “proteasome bounce back response”, which is the transcriptional activation of proteasome subunit genes in response to proteasome inhibitors. Importantly, NRF1 is a key regulator of the proteasome bounce back response. NRF1 inhibition sensitizes cancer cells to proteasome inhibitors, suggesting that the suppression of NRF1 activity may have a synergizing effect with proteasome inhibitors for the treatment of cancer patients exhibiting elevated proteasome activities [25].

### 5.7. Combination of NRF1 Inhibitors and Proteasome Inhibitors

As expected, the suppression of NGLY1, a regulator of NRF1 activation, attenuates NRF1 accumulation and increases cell death in cancer cells exposed to proteasome inhibitors [78]. Similarly, disruption of the TIP60 complex, which is required for NRF1 transcriptional activity, leads to the decreased viability of proteasome inhibitor-treated cancer cells [85]. The effectiveness of the combination of NRF1 inhibition and proteasome inhibition was verified in cell culture experiments. However, whether this combination truly affects in vivo tumorigenesis remains to be tested.

### 5.8. Clinical Correlation between OGT and Proteasome Subunit Proteins

Recently, the suppression of OGT was demonstrated to enhance the efficacy of proteasome inhibitors against solid tumors in xenograft mouse models [28]. As high OGT activity is correlated with poor clinical outcomes in cancer patients, OGT is anticipated to be a target molecule for cancer therapies [68]. Although no apparent inhibition of tumor growth was observed by OGT suppression alone or proteasome inhibitor treatment alone in a lung cancer xenograft model, the combination of these two significantly reduced tumor growth [28].

Intriguingly, datasets deposited in The Cancer Genome Atlas (TCGA) revealed a positive correlation between proteasome subunit expression levels and OGT levels, which is regarded as an NRF1 activator, in breast cancer cases [28]. Thus, as a regulator of NRF1, OGT is a promising target for use in combination with proteasome inhibitors as an anti-cancer therapy.

### 5.9. Roles of Other CNC Family Transcription Factors in Cancer

NRF3 has been identified as another bZIP–CNC transcription family member [86]. Whereas NRF3 possesses an ER-binding region that is identical to that of NRF1 and is regulated in a similar manner as NRF1, NRF3 directly upregulates a gene encoding the 20S proteasome assembly chaperone, the proteasome maturation protein (POMP), but it does not activate proteasome subunit genes, which is distinct from NRF1 [87,88,89]. Activation of the 20S proteasome subunits by NRF3 confers growth advantages on colon cancers through the degradation of the p53 protein [89]. As nuclear NRF3 is degraded through the β-TrCP E3 ligase, which is released from NRF1 upon its *O*-GlcNAcylation, it is likely that NRF3 is *O*-GlcNAcylated and stabilized in the nucleus, similar to NRF1. It remains unclear whether and how NRF1 and NRF3 regulate proteasome activity cooperatively in cancer cells.

## 6. Concluding Remarks 

Many cancer cells exhibit increased levels of glucose and glutamine uptake, leading to an abundant synthesis of UDP-GlcNAc that fuels OGT-mediated *O*-GlcNAcylation and the subsequent accumulation of NRF1 protein and the facilitation of the proteasome pathway (Figure 5). On the other hand, NRF2 activates the pentose phosphate pathway, nucleotide synthesis, and glutathione synthesis by utilizing glucose and glutamine (Figure 5). As both proteasome subunit genes and NRF2-regulating metabolic pathway genes contain ARE sequences, the functional differences between NRF1 and NRF2 raise the question of whether these two CNC factors compete with each other to activate these target genes; in fact, it has been reported that NRF1 competes with NRF2 at the ARE sequence in the *Slc7a11* gene in mouse liver [90]. This study demonstrated that one of the NRF2 target genes, *Slc7a11*, was upregulated in NRF1-deficient mouse liver, which was interpreted as that NRF1 and NRF2 represses and activates *Slc7a11*, respectively, in a competitive manner because NRF1 and NRF2 share the consensus DNA binding motif.

In the case of proteasome subunit gene regulation, NRF1 is regarded as a key regulator, allowing the hyperactivation of the proteasome in several cancer species through the transcriptional activation of proteasome subunit genes because NRF1 has been shown to make a major contribution to the induction of proteasome subunit genes in response to proteasome inhibitors. Supporting this interpretation, a significant correlation between OGT, an NRF1 activator, and proteasome subunit genes was observed in clinical breast cancer specimens [28]. In contrast, NRF2 was reported to transcriptionally activate the expression of proteasome subunit genes in a few cancers, including colon and breast cancer [91,92], and it was also demonstrated to be an activator of proteasome subunit genes in cells treated with electrophiles, such as sulforaphane and 3H-1,2-dithiole-3-thione [93]. Both NRF1-sMAF and NRF2-sMAF heterodimers have been shown to bind to the ARE sequence in the promoters of proteasome subunit genes [24,25,81,82,91,92,93]. According to these reports, NRF1 and NRF2 share target genes, and both activate their transcription but in different cellular contexts, namely, in response to distinct endogenous or exogenous signals.

Focusing on NRF1, we consider that metabolism impacts proteostasis in cancer cells. NRF1 transduces intracellular glucose and glutamine levels to induce proteasomal activity via the *O*-GlcNAcylation of NRF1 consuming UDP-GlcNAc as a substrate. The *O*-GlcNAcylation sites of NRF1 overlap with GSK-3β-dependent phosphorylated serine residues required for the interaction with β-TrCP [28,41]. GSK-3β is reported to phosphorylate not only NRF1 but also NRF2 and NRF3, leading to β-TrCP proteasomal degradation [41,42,87]. This similarity raises the question of whether increased cellular *O*-GlcNAcylation also enhances the protein levels of NRF2 and NRF3. Notably, KEAP1 is reported to be *O*-GlcNAcylated, which elevates KEAP1 E3 ligase activity and promotes the degradation of NRF2 in cells with increased cellular *O*-GlcNAcylation [94], suggesting that NRF2 is not involved in the activation of proteasome subunit genes in cancer cells with high *O*-GlcNAcylation activity. Thus, cellular UDP-GlcNAc and *O*-GlcNAcylation levels are most likely to determine transcription factor specificity for proteasome subunit gene regulation: NRF1 or NRF2. Interestingly, oxidative stress evokes an unfolded protein response that activates the NRF1 pathway, and moreover, cellular *O*-GlcNAcylated proteins are increased in response to many stresses, including oxidative stress [95,96]. Cross-talk between oxidative stress and proteotoxic stress (unfolded protein stress) factors in various nutrient states might lead to optimal response modes mediated by CNC transcription factors for cytoprotection.

*C. elegans* is a well-established animal model used to study how CNC transcription factors respond to different stresses because *C. elegans* possesses only a single gene, skn-1, which encodes NRF1 and NRF2 orthologs [97]. Skn-1 regulates both proteasome subunit genes and antioxidant/detoxification genes in response to each corresponding stimulus, which is mainly regulated by phosphorylation signaling [98]. Notably, skn-1 is *O*-GlcNAcylated [99], and the overexpression of GFAT, which is a rate-limiting enzyme of UDP-GlcNAc synthesis, confers resistance against ER stresses in *C. elegans*, suggesting that UDP-GlcNAc is a critical metabolite for protein homeostasis [100,101]. A close relationship between metabolism and proteostasis via NRF1 *O*-GlcNAcylation is a highly conserved defense mechanism among various species that is hijacked by cancer cells.

## Figures and Tables

**Figure 1 cancers-13-00541-f001:**
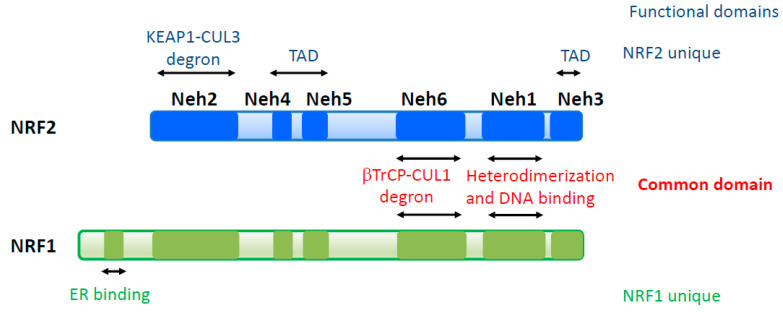
Structure of NFE2L2 (NRF2) and NFE2L1 (NRF1) transcription factors. NRF2 and NRF1 share two functionally conserved domains involving -TrCP-dependent proteasomal degradation (Neh6) and DNA binding and heterodimerization with small MAF (sMAF) (Neh1). NRF2 possesses unique domains involving transactivation (Neh4/5) and Keap1-dependent proteasomal degradation (Neh2). NRF1 contains a unique endoplasmic reticulum (ER)-binding domain at its N-terminal region. TAD, transactivation domain; DBD, DNA binding domain.

**Figure 2 cancers-13-00541-f002:**
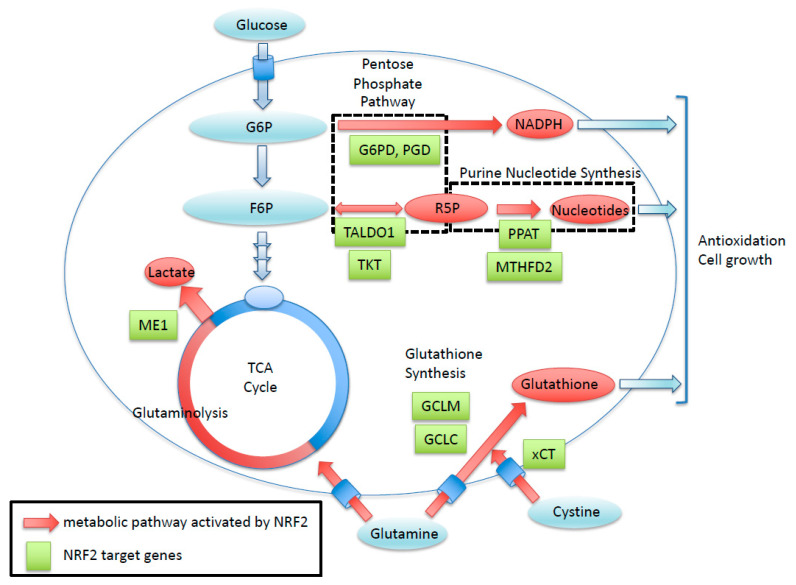
Roles of NRF2 in cancer metabolism. NRF2 enhances the cancer metabolic pathway through the gene expression involving the pentose phosphate pathway, purine nucleotide synthesis, glutaminolysis, and glutathione synthesis. Green boxes exhibit NRF2-target genes with antioxidant response element (ARE) sequence. Red arrows show metabolic pathways activated by NRF2. R5P, Ribose 5-phosphate; G6P, Glucose-6-phosphate; F6P, Fructose 6-phosphate.

**Figure 3 cancers-13-00541-f003:**
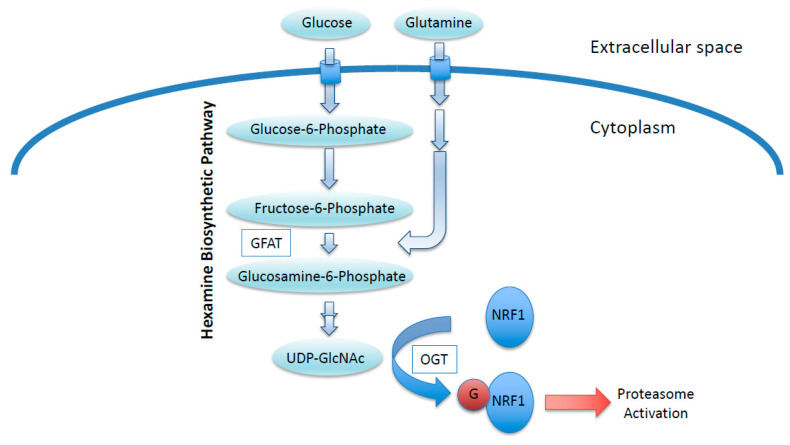
NRF1 stabilization regulated by an oncometabolite UDP-GlcNAc. Increased glucose and glutamine uptake in cancer cells increase cellular UDP-GlcNAc levels, leading to increased O-GlcNAcylation of NRF1. O-GlcNAcylation antagonizes the proteasomal degradation of NRF1, and stabilized NRF1 activates the transcription of proteasome subunit genes. GFAT, Glucosamine:fructose-6-phosphate aminotransferase; OGT, O-GlcNAc transferase; G, O-GlcNAc moiety.

**Figure 4 cancers-13-00541-f004:**
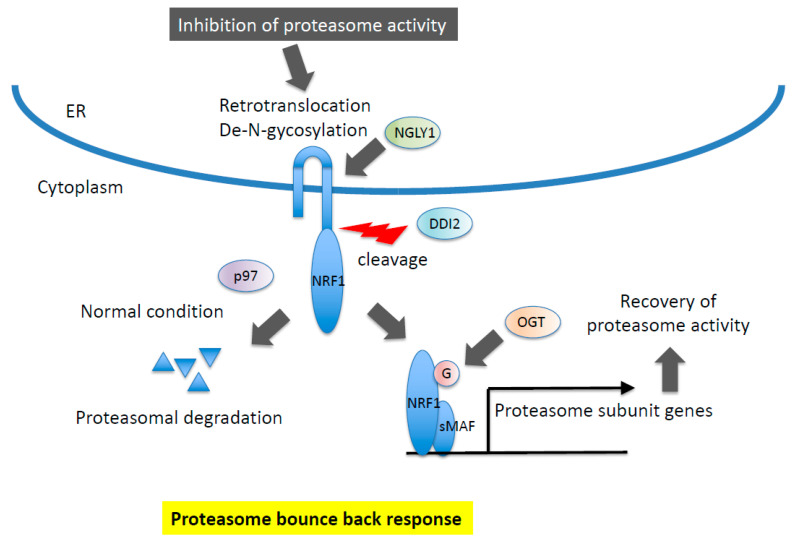
Molecular mechanism of NRF1 protein regulation. NRF1 is subjected to proteasomal degradation in ER (ER-associated degradation, ERAD) under normal conditions. In response to inhibition of proteasome activity, NRF1 is retrotranslocated and de-N-glycosylated by p97 and NGLY1, respectively. DDI2 protease recognizes the DSGA motif in NRF1 cytosolic region and cleaves. Processed NRF1 is translocated in the nucleus and O-GlcNAcylated, resulting in the transcriptional activation of target genes including proteasome subunit genes.

**Figure 5 cancers-13-00541-f005:**
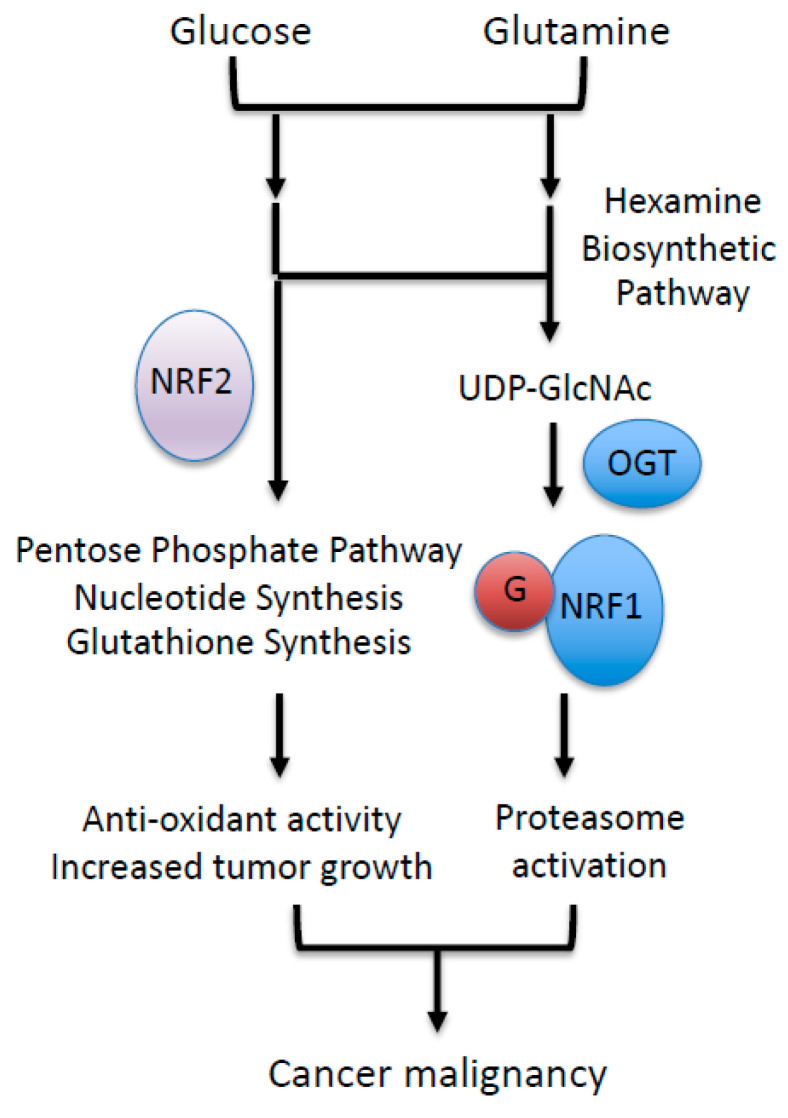
CNC (cap‘n’collar) transcription factors and glucose/glutamine metabolic pathway. NRF2 is an upstream regulator of glucose and glutamine metabolism via the activation of genes involved in the pentose phosphate pathway, purine nucleotide synthesis, and glutathione synthesis. NRF1 is activated downstream of glucose and glutamine metabolic pathway via its increased O-GlcNAcylation.

## Data Availability

No new data were created or analyzed in this study. Data sharing is not applicable to this article.

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
