# Peer review of "Roles of CNC Transcription Factors NRF1 and NRF2 in Cancer"

_cancers, 2021, doi:10.3390/cancers13030541_

Round 1
Reviewer 1 Report
The authors have adequately addressed my points. I have no further comments.
Reviewer 2 Report
The authors have satisfactorily addressed the concerns.
However, the legend for figure 2 is misplaced in the paper (lines 98 to 102), please check.
The grammatical error was corrected in the revised edition.
This manuscript is a resubmission of an earlier submission. The following is a list of the peer review reports and author responses from that submission.
Round 1
Reviewer 1 Report
The NRF proteins are recognized for their cytoprotective and antioxidant properties with NRF2 being the most studied and which is known to regulate metabolic pathways in cancer cells. Cancer cells exhibit unique metabolic alterations that ensure their survival and proliferation in nutrient-deprived environment. In this review the authors describe the interplay of 2 NRF proteins, NRF1 and NRF2 in cancer metabolism. While the review does provide insights into the divergent functions of NRF1 and NRF2 in cancer cells, the review lacks critical analysis of the literature.
Following are the suggestions to improvise the review.
Line 22: Avoid repetitive use of “mediated”, maybe you could replace mediated with “effected by cancer cell metabolism”
Section4: The function of NRF2 in metabolic reprogramming of cancer cells could be elaborated further in the text as well as a figure illustrating NRF2-regulated metabolic pathways in cancer cells. While figure 4 does provide a brief overview of NRF2 function, a more detailed figure depicting regulation of genes/metabolites by NRF2 and concomitant contribution to cancer cell survival and growth would be appropriate.
Section 5 could be sequenced better to maintain focus. Start with 5.3 then 5.4 omit 5.2(not relevant) and then 5.1
Line 289: Please correct the grammatical error.
Reviewer 2 Report
This review summarizes the present knowledge of NRF1, a transcription factor and a member of the CNC family, with particular emphasis on its role in the regulation of proteasome activity. The authors show an excellent expertise in the topic of the review.
This work can be further improved by introducing a few changes.
1) In general, the chapters result very concise and do not clearly convey the message. A review is not necessarily for experts. Thus, the reader would greatly benefit from a more detailed description of the statements. The authors should make some extra effort to expand the concepts and make them more available to the readers.
2) Most of the content of this review is on the relationship between NRF1 and proteasome bounce back response, while metabolism intended as metabolic reprogramming, such as that commonly occurring in cancer cells, is very little discussed. The authors should consider to modify the title (i.e. inserting proteasome) and expand chapters 4 and 5, particularly 5.2.
3) Concluding Remarks: “In this review, we describe the relationship of NRF1 and NRF2 with cancer metabolism”. This is an overstatement. The authors have not really described the relationship of the two transcription factors with cancer metabolism. Please delete or change this sentence.
4) The paragraph concerning the competition between NRF1 and NRF2 at the ARE sequence is unclear and should be better detailed.
5) Along this line, it would be useful to achieve some more information about the differences and similarities between NRF1 and NRF3, the other member of the same CNC family.
Minor comments:
Please, spell out the several abbreviations present in the Abstract.
Some words and symbols are missing throughout the text (i.e. lines 207 and 289).